# Emotional Experiences of COVID-19 Patients in China: A Qualitative Study

**DOI:** 10.3390/ijerph19159491

**Published:** 2022-08-02

**Authors:** Yu Deng, Huimin Li, Minjun Park

**Affiliations:** 1College of Language Intelligence, Sichuan International Studies University, Chongqing 400031, China; 2School of English, Sichuan International Studies University, Chongqing 400031, China; 10201901120160@stu.sisu.edu.cn; 3Chinese Language and Literature, Duksung Women’s University, Seoul 01369, Korea

**Keywords:** COVID-19 patients, China, emotional experiences, mental health, qualitative analysis

## Abstract

This study explored the emotional experiences of COVID-19 patients in China. Thirty-four patients diagnosed with COVID-19 participated in semi-structured telephone interviews. We used qualitative methods to investigate the distribution patterns and characteristics of patients’ emotional experiences. The results indicated that emotional experiences showed different characteristics at different stages during isolation and treatment. COVID-19 patients’ emotional discourse encompassed eight main themes, namely, feelings of shock at the diagnosis, yearning for future life, attachment to one’s family, depression during the treatment, self-restriction due to probable contagiousness, powerlessness about the disease, open-mindedness about death, and faith in the joint efforts to fight COVID-19. These themes related to experiences concerning infection, isolation, outlook on life and death, stigma, and macro-identity. The findings suggest that the unexpected experience of COVID-19 infection exacerbated patients’ negative emotions. COVID-19 patients’ emotional stress stemmed from isolated environments, physiological effects of the disease, panic about the unknown, and realistic economic pressure. The government, medical staff, family members of patients, and the media should therefore work together to ensure proper emotional care for COVID-19 patients.

## 1. Introduction

Quality of life is strongly associated with our health status, which means not only the absence of illness, but also a state of physical, mental, and social well-being [1,2,3]. Thus, the assessment of quality of life is linked with physical health, psychological state, social relationships, personal belief, and our relationship to the living environment. Health-related quality of life has been extensively explored in different medical disorders such as heart disease, HIV, lung cancer, and SARS [4,5,6,7]. The present study aimed to explore health-related quality of life among COVID-19 patients in China, and its domain specific to emotional experiences regarding COVID-19 infection.

Since the outbreak of the COVID-19 pandemic, an exponential growth of COVID-19-related studies have reported that many individuals experienced negative emotional events due to fears of contagion and family members’ deaths [8]. As a traumatic event, COVID-19 infection has led to various psychological disorders among patients, such as PTSD (post-traumatic stress disorder), anxiety, and depression [9,10,11,12,13]. COVID-19 patients have been especially vulnerable to emotional turmoil during the pandemic.

A considerable number of studies have investigated the psychological experience or lived experiences of COVID-19 patients during or after their hospitalization [9,10,14,15,16,17,18]. Jesmi et al. [19] reported that a large proportion of COVID-19 patients expressed the fear of death. This corroborates Sun et al.’s findings that COVID-19 patients felt uneasy and uncertain about the results of medical examinations during their hospitalization [15]. Similarly, a study showed that COVID-19 patients who were quarantined believed that their social relations were hampered by physical restrictions given that they might carry the virus [20]. Some studies emphasized that the disclosure of personal information during quarantine posed a risk to COVID-19 patients’ mental health [15,18,21].

Fear is the breeding ground for hatred and stigma. Social stigma has been uncovered among most COVID-19 patients. It is vital to avoid stigma regarding COVID-19 infection as stigma can make individuals hide their illness and not seek healthcare immediately. Stigmatized experiences and fears of being stigmatized pose major mental health risks to COVID-19 patients, sometimes leading to extreme life decisions [22]. One explanation is that due to limited knowledge about the virus, infection, and treatment, individuals tend to blame the spread of the virus on others’ negligence [22]. Another explanation is that isolation, as a widely recognized pandemic prevention measure, inevitably stigmatizes patients as “others” in the hospital, as opposed to “us” outside the hospital [23].

In China, patients with a suspected or confirmed case of COVID-19 also exhibited high levels of psychological stress. A survey of 714 clinically stable COVID-19 patients in Wuhan showed that the prevalence of major post-traumatic stress symptoms associated with COVID-19 was as high as 96.2% [24]. Another survey questionnaire of PTSD, depression, and anxiety scales with 126 quarantined COVID-19 patients in Shenzhen also demonstrated a high rate of psychological distress among the COVID-19 survivors in the early recovery stage [25]. Using a qualitative narrative analysis method, Wu et al. [18] investigated the health-related quality of life of 16 hospitalized COVID-19 survivors in Nanning city. The narrative analysis showed that COVID-19 patients suffered from anxiety, trauma, and self-stigma. Patients’ negative emotions were linked with abnormal social interactions. Drawing on a thematic analysis of narratives of thirteen COVID-19 patients in Wuhan, Li et al. [26] found that patients experienced negative emotions such as confusion, uncertainty, worry, and guilt due to social discrimination and poor financial security. Patients’ positive emotions were concerned with expectations about making up for lost time with family after discharge. Most recently, Sun et al. [15] conducted semi-structured interviews with 16 individuals who were among the first diagnosed and treated in the COVID-19 pandemic. The results demonstrated the diachronic change rule of COVID-19 patients’ emotional potency. Specifically, upon learning of their infection, the first emotions that emerged were denial, shame, and fear; after a buffer period, patients gradually accepted and faced the medical treatment in the hospital, and in the later stages, patients’ emotions became even more complicated, as they were uneasy about the virus, but they had begun to have hope for promising examination results. Clearly, narrative discourse can be an indicator of emotional disorders and represents a new research trend with respect to COVID-19 patients’ mental health [15].

Although there is a growing body of research on the emotional state and mental health issues related to COVID-19 infection, the primary methodology has been the utilization of self-report questionnaires at one timepoint, which cannot provide information on patients’ emotional changes. To date, there have been several broad qualitative studies concerning COVID-19 patients’ mental health in China [9,10,15,18,26], but there is a lack of in-depth studies examining the emotional discourse of patients with confirmed cases of COVID-19.

In the present study, we interviewed 34 individuals infected with COVID-19 in China to investigate their emotional discourse. The phenomenological qualitative interview method can offer insights into patients’ first-person perspectives and illuminate how their quality of life is affected by the COVID-19 illness in an in-depth way [14]. Furthermore, the phenomenological observation of rich emotional experiences of COVID-19 patients in China should be of interest to a broad worldwide readership within health care and may inform the measures taken worldwide to cope with mental health issues regarding COVID-19 infection [26]. Given the uniqueness, complexity, and multidimensional nature of COVID-19 patients’ life experiences and characteristics, qualitative emotional discourse analysis based on in-depth interviews is an appropriate way to provide a deeper understanding of COVID-19 patients’ emotions and psychological experience [27]. We hope the findings can serve to determine the prevalence of certain emotional tendencies in COVID-19 patients and to identify relevant interventions to improve their well-being.

The conceptual framework of the present study is shown in Figure 1. The emotional experiences of COVID-19 patients were explored systematically in terms of narrative discourse analysis, by combining the five basic emotions (i.e., happy, angry, sad, fear, disgust) [28] and 21 subcategories of emotions (i.e., joy, comfort, respect, praise, trust, like, wish, angry, upset, disappointed, guilty, grief, panic, dread, shame, depressed, hate, criticize, envious, suspect, surprise) [29]. Based on the thematic analysis of emotional narratives, mental health problems of COVID-19 patients were identified in line with the underlying risk factors or causes. Lastly, we proposed potential solutions for intervention and management concerning COVID-19 infection.

## 2. Material and Methods

### 2.1. Research Design

This study employed one-hour, semi-structured telephone interviews with 34 Chinese COVID-19 patients from September 2020 to March 2021 to collect narrative discourse data. (The present study is a sub-project of the research project “COVID-19 patient narratives and their mental health.”). The transcribed narrative data were coded to analyze patients’ emotions and psychological status.

### 2.2. Participants

We recruited the 34 participants from the Chongqing Public Health Medical Center through convenience sampling. The inclusion criteria were a confirmed diagnosis of COVID-19 and a demonstration of the clinical symptoms of the COVID-19 disease. Furthermore, participants had been in a hospital quarantine and received outpatient or inpatient care. Participants included 16 women (47%) and 18 men (53%). The mean age was 47 years. All patients had completed at least nine years of compulsory education. Twenty-four participants were married, eight were single, and two were divorced. Sixteen were employed, eleven were self-employed, two were students, one was retired, and four were unemployed (one housewife). In terms of length of stay, 24 patients had been hospitalized for over 10 days, and 6 had been hospitalized for over 30 days (Table 1). Prior to the interviews, participants signed a form to indicate their voluntary participation in the study. Patients received a cash reward for their participation. The study was conducted in accordance with the Declaration of Helsinki and approved by the Ethics Committee of Chongqing Public Health Medical Center (Approval Date: 4 September 2020; Approval Number: 2020-048-02-KY).

### 2.3. Interview Procedure

Patients were contacted by telephone or WeChat. The semi-structured telephone interviews were audio-recorded with the participants’ consent. Each interview lasted approximately 60 min, depending on the patient’s condition. If patients were unwilling or unable to continue the interview, the interview was terminated immediately. Additionally, we collected demographic information from patients through WeChat.

Guided by the interviewer, patients spoke freely about what they had experienced during the pandemic and their emotions concerning their COVID-19 infection. Interview questions included the following: “How were you infected with COVID-19?”; “How did you feel when you were diagnosed with COVID-19?”; “What did you do before, during, and after hospitalization?”; “How did you feel during the hospital treatment?”; “How did healthcare staff help you in the hospital?”; “How were your work and daily life affected by the infection?”; “What did you do when you felt frustrated?”; “What changes in your attitude toward life and death did you have?”; “How did you deal with social relationships after discharge?”, and “what was the most unforgettable event in your recovery process?” [9,10,11]. The interview questions were semi-structured. Each patient answered all the questions by freely expressing their personal experience and feelings. The order of the questions asked was flexibly adjusted according to the specific situation during the interviews [26]. The interviews were open-ended to elicit participants’ lived experiences and emphasized listening to the participants [14]. After all interviews were completed, the interviewer organized the project team to transcribe the interview audio files. We used the NVivo software to support emotional discourse analysis and content extraction from the transcribed corpus.

### 2.4. Emotional Narrative Tagging and Theme Extraction

The study excluded the interviewer’s texts and retained the interviewees’ narrative texts. For emotional narrative tagging, we combined the 21 subcategories of emotions outlined by Xu et al. [28] and the five basic emotions (happy, angry, sad, fear, disgust) from Hong et al. [29]. The early works of Ekman [30,31] provided the theoretical foundation for the 5/21 emotional framework in the present study. Ekman classified the six basic human emotions as anger, disgust, fear, happiness, sadness, and surprise, noting that in one preliterate culture fear and surprise were not distinguished from each other although both were differentiated from anger, sadness, disgust, and happiness [32]. Resting on Ekman’s pioneering emotional classification, Xu et al. [28] developed a Chinese affective lexicon ontology for natural language processing in light of seven basic emotions (happy, good, angry, sad, fear, disgust, and surprise) and 21 emotional subcategories (see Table 2). Recently, Hong et al. [29] optimized Ekman’s classification as five basic types—happy, angry, sad, fear, and disgust—in a narrative analysis of persons with schizophrenia. The present study integrated Hong et al.’s [29] five basic emotions and Xu et al.’s [28] twenty-one subcategories of emotions in order to involve a wide range of social and psychological experiences of COVID-19 patients. Table 2 shows the modified emotional framework for narrative discourse analysis. Specifically, we classified Xu et al.’s [28] subcategories *Respect (PD)*, *Praise (PH)*, *trust (PG)*, *Like (PB)*, and *Wish (PK)* into “HAPPY” rather than “Good”, *Depressed (NE)* and *Suspect (NL)* into “FEAR” rather than “Disgust”, and *Surprise (PC)* into either “HAPPY” or “FEAR” according to the context. It is noteworthy the capitalized two-letter tags (e.g., PA, PE, NE, NC) were set for ID tagging, presumably P for positive and N for negative [28].

Emotional narrative analysis was conducted by three independent coders. Table 3 illustrates examples of emotional narrative tagging. The ID tags quoted in parentheses concern secondary emotional classes due to their brevity and identifiability. All primary and secondary emotional tags were enclosed within special tokens ‘++’ for the automatic identification of sentiment in the structured narratives dataset during the text processing procedure. It should be noted that positive polarity (i.e., HAPPY) and negative polarity (i.e., ANGRY, SAD, FEAR, DISGUST) were not proportional among the observed five primary emotional prototypes (i.e., HAPPY, ANGRY, SAD, FEAR, DISGUST). To avoid bias toward negative emotions, we augmented the typical HAPPY class with several emotional subcategories such as *Respect*, *Praise*, *Trust*, *Like*, and *Wish* (see Table 2). This measurement appeared to increase narratives of the HAPPY emotion but did not directly affect theme extraction given that the themes were derived from the comprehensive utilization of the 21 subcategories and discourse features of the texts rather than the five primary emotional categories alone.

After emotional narrative tagging, we identified several main themes and subthemes of the emotional discourse related to the lived experiences of COVID-19 patients based on the structural reading and interpretation in line with the machine clustering method implemented in NVivo. The process of deriving units of significance and themes followed the broad thematic analysis approach [33]. First, three coders read and re-read the interview texts in line with the interview questions and conducted a manual classification of the emotional content (see Table 3). Second, the texts were inputted into NVivo, and initial codes were generated for the five basic emotions and 21 emotional subcategories. Sentence segments of the interview texts were mapped onto each emotional category in a systematic fashion. Third, based on the preliminary manual emotional classification in step one, the three coders read the interview texts in depth again, and sorted out the emotions in the narrative discourse for the second time. Fourth, after the coding value for each emotional paragraph was assigned, the “hierarchical chart” function in NVivo was used to generate the distribution of emotional labels across patient narratives. Then, the “cluster analysis” in NVivo was implemented based on “word similarity” and “Pearson correlation coefficient” to generate emotional code clusters. Fifth, the three coders reviewed the interview questions again and collated emotional codes into preliminary themes based on the results of step four. When different levels of content arose for a theme, the theme was divided into specific subthemes. The three coders checked whether the themes worked in relation to the coded extracts and the entire data set, and then generated a thematic map. Finally, the “text search” function in NVivo was employed to find relevant paragraphs in each interview text for the themes obtained in step five. The three coders read the paragraphs concerning the relevant themes and refined the specifics of each theme. Disagreements regarding emergent themes and subthemes were resolved through discussions among the three coders.

## 3. Results

The emotional narrative analysis detected eight main themes and nine subthemes related to COVID-19 patients’ emotions, as shown in Figure 2. The eight main themes included sense of shock at the diagnosis, hope for future life, attachment to one’s family, depression during treatment, self-restriction due to probable contagiousness, powerlessness about the disease, open-mindedness about death, and faith in the joint efforts to fight COVID-19. Among the eight major emotional themes, depression during treatment and faith in the joint efforts to fight COVID-19 were the two most prominent. The sense of depression during treatment can be subdivided into four subthemes, namely humiliation at being treated unfairly, resentment about the pandemic’s negative consequences, bad luck/misfortune and suffering, and discomfort with physiological symptoms. Patients’ faith in the joint efforts to fight COVID-19 can be divided into five subthemes, namely mutual support between patients, an inclusive and considerate social environment, respect and tender love for medical staff, trust and love for the country, and positive feelings of giving back to society.

### 3.1. Sense of Shock about the Diagnosis

At the time of patients’ COVID-19 diagnosis, the first emotion that emerged was doubt. They claimed that they had been as careful as possible in all aspects to avoid exposure to the virus. They were surprised and disappointed that the various measures that they had taken had failed. They constantly recalled the various situations of possible exposure to COVID-19, attempting to determine an exact time. Generally, patients felt shocked when their COVID-19 infection was confirmed. Several examples follow:

(1) *a. Patient 14: ++FEAR(NL)++ At that time, I was wondering whether the hospital had made a mistake. It felt unlikely. Because I was in good health, how could I get infected with COVID-19? It’s weird.*


*b. Patient 25: ++FEAR(NL/NI/PC)++ Was there an error in the COVID-19 test? Then, another test on the same day showed negative, but it tested positive on the second day; there were antibodies…that is, when I first saw them, I didn’t understand. It indicated igM and igG, which suggested positive. I was completely confused at that time.*



*c. Patient 4: ++FEAR (NE/NL)++ So, I had been thinking about this problem. I said, “How was I infected with COVID-19 this time?” So, I had been thinking about it. Was it due to the failure of disinfection in the house, or in the hotel, or in the air?*



*d. Patient 29: ++FEAR (PC)++ I was shocked and surprised at that time because when I boarded the flight, I wore protective clothing, and I also wore an N95 mask. I even wore protective glasses, and I didn’t drink water on the flight. So, I didn’t know how I was infected with COVID-19.*


Excerpts (1a–d) illustrate that the patients were unaccepting and skeptical in the initial stage of COVID-19 infection. Again, the first emotions that emerged were surprise, denial, and panic [15]. In this stage, patients forced themselves to recall how they had contracted COVID-19. The focal negative emotion during the initial stage was fear in that patients did not fully accept the fact that COVID-19 could infect them and lacked scientific knowledge of COVID-19 and the treatment procedures. Hence, individuals around COVID-19 patients should show kindness to reduce patients’ initial fear.

### 3.2. Hope for Future Life

After a buffer period, COVID-19 patients were gradually willing to accept their diagnosis and receive treatment in the hospital. At this stage, COVID-19 patients experienced conflicting emotions. On the one hand, they were still uneasy about the virus; on the other hand, they had become hopeful for favorable examination results [15]. See the following examples:

(2) *a. Patient 33: ++HAPPY(PK/PE)++ I’m going to actively cooperate with the treatment so as to leave the hospital as soon as possible.*


*b. Patient 15: ++HAPPY(PA)++ It might be the first day or the second day of hospitalization. I remember that it was always gloomy in Chongqing at that time, but the next day, I seemed to see the sunshine, which shone through the fog in Gele Mountains, and then penetrating through the windows. I felt hopeful at that moment. The weather was very good on the day of my hospital discharge.*



*c. Patient 6: ++SAD (PF/NH)++/++HAPPY(PA)++ At that time, I thought it was the beginning of my infection, and I didn’t know whether COVID-19 could be cured, something worse might happen, or whether some consequences might occur. I felt if I could recover from COVID-19 or got well, I would live my life seriously and happily after returning home, in particular getting along well with my family. If I want something, I will try to do it and try to have it. I must not just muddle along my life day by day; I just have that feeling.*



*d. Patient 20: ++HAPPY (PH/PK)++ I felt touched in my heart, and I felt if I had…ability, if I had ability, I want to repay society. But, considering my age and my personal ability, there is no such possibility. I just felt that in the future, if there is an afterlife, I must work hard and be a…a person of value. Right…well, just like the medical staff, I want to do something for the country.*


Excerpts (2a–d) demonstrate COVID-19 patients’ conflicting emotions. On the one hand, the COVID-19 treatment process interrupted patients’ original life trajectory, leading to months of solitude and thinking. Sadness arose during patients’ long-term hospitalization. On the other hand, during this lengthy isolation, many patients reevaluated their self-worth and reflected on their past life. The experience of a major illness or even a near-death experience showed patients the impermanence of life. People generally believe that “life will always exist.” When this subconscious belief is challenged, patients must reconsider “what really matters.” As such, COVID-19 patients reshaped their mentality with a definite idea of “starting a new life after the end.” There was still some uncertainty during this waiting period, and patients were not sure whether they would experience social pressure from others. Thus, patients demonstrated mixed emotions of grief, guilt, and hopefulness. Overall, however, patients paid more attention to reassessing their lives and emotionally expected a kind of perfection and happiness contrasting with their current isolation.

### 3.3. Attachment to One’s Family during Hospital Isolation

Weeks or months of social isolation in the hospital made patients feel lonely, which was magnified by the anxiety and vulnerability caused by waves of physical and mental symptoms. Patients exhibited complicated emotions regarding family reunion. While they missed their families, they felt reluctant to impose burdens on them.

(3) *a. Patient 34: ++SAD(NB/PF)++ I definitely miss my family, and it was about time for the New Year. With this unlucky infection of COVID-19, I must spend the New Year here in the hospital.*

*b. Patient 29: ++FEAR(NE)++/++SAD(NJ)++The Spring Festival was coming soon, wasn’t it? My family was looking forward to my early return, but I was here in the hospital and couldn’t go back. I wanted to go back home soon, but the situation did not permit. My mood would become worse, alas. I felt very sad and very depressed at that time*.

Excerpts (3a–b) describe patients’ sadness and depression during their isolation in the hospital. The COVID-19 outbreak started during the Chinese New Year, and many patients were either on the way to reunite with their family or had already visited their family. Given the COVID-19 infection, patients had to leave their families and loved ones for lengthy treatment and quarantine during the festival of reunion. The Spring Festival made patients’ loneliness particularly strong. Consequently, patients’ negative emotions such as grief and disappointment emerged during the Chinese New Year, as they were eagerly looking forward to reuniting with their loved ones.

To reduce the worries of families, COVID-19 patients tended to be secretive about their infection, as narrated by two patients who were hospitalized during illness:

(4) *a. Patient 6: ++FEAR(NC/NE)++ I didn’t tell my family about the COVID-19 infection. I didn’t dare to tell my family. My deepest feeling was that I was very afraid to have them worry about me. My mother’s health was not fine, and I was afraid that she would worry all day. Hence, I just didn’t want my family to know about my illness.*


*b. Patient 33: ++FEAR (NC)++ At present, I have to hide my infection from my family. After all, my family was very afraid of this virus subconsciously, but they did not know about my current situation. Actually, there’s nothing to worry because the main treatment was as common as taking Chinese medicine and getting a regular COVID-19 test, and nothing else. We didn’t feel a big burden in our hearts when we were treated in the hospital, and it’s not a big problem. However, I dared not tell our families about my COVID-19 disease for fear that they would worry about me. I would tell them a little after leaving the hospital. I didn’t consider this problem at present.*


As excerpts (4a–b) show, patients felt reluctant to inform their families of their health status so that they could protect them, with some not even telling their families that they had been diagnosed with COVID-19 or quarantined. Some did not keep their isolation a secret, but they avoided mentioning the progression of the illness and their physical discomfort to their families. In patients’ minds, there was no need to let the family suffer in vain. Consequently, patients’ fear, depression and loneliness increased with the uncertainty of COVID-19 infection and reinfection. Their emotional states deteriorated with the absence of connection with families.

### 3.4. Depression during Treatment

#### 3.4.1. Humiliation from Being Treated Unfairly

Control of the COVID-19 pandemic is a social matter, which means that patients receive close attention from individuals and various social institutions. COVID-19 patients can be treated unkindly, and our participants reported experiences of discrimination by social institutions. If their private information was leaked, it would cause great harm to patients’ daily life and mental health. Under intense scrutiny by the outside world, patients experienced a heavy psychological burden. Some lost their normal social status, going from “ordinary people” to “people controlled by ordinary people.” Angry and disappointed, COVID-19 patients wanted to be treated fairly.

(5) *a. Patient 3: ++ANGRY (NA) ++/++SAD (NB)++ When my sister’s information was exposed, it was the most unhappy time. Because it is extremely upsetting to expose her information, I felt very angry about that.*


*b. Patient 24: ++DISGUST (NN)++ I think this is not my fault that I contracted COVID-19. Some institutions or people, first of all, should make it clear that we are not prisoners but victims. Some people look at COVID-19-infected persons with strange eyes and strange moods.*



*c. Patient 23: ++SAD (NB)++ Um…I still felt like I hadn’t been cured. I felt no freedom before I completely recovered from the COVID-19 disease. Although the food was very good, I had the feeling of having made a mistake. I even felt like a prisoner.*



*d. Patient 31: ++SAD (NJ)++ What I wanted was to be treated equally.*


Excerpt (5a) shows that some patients’ private information was leaked, which led to anger and sadness among patients and their families. Individuals discussed COVID-19 patients’ life experiences on the internet and made negative comments, disrupting patients’ everyday lives, and hurting their feelings greatly. Patient 3 mentioned that the individuals who had exposed his information had apologized to him and to his family, which made them feel much better. Excerpts (5b–c) demonstrate COVID-19 patients’ emotions of disgust and sadness when treated unfairly. Even after they had recovered and been discharged from the hospital, patients were still stigmatized. In their account, they were not only regarded as “people who are ill” but also as “people who are still dangerous to others.” This unfortunate experience exacerbated patients’ negative emotions and psychological problems. Excerpt (5d) summarizes the desire of all COVID-19 patients to be respected and receive appropriate, humane treatment when they felt disappointed with unfair treatment.

#### 3.4.2. Resentment about the Pandemic’s Negative Consequences

While undergoing treatment, COVID-19 patients experienced complicated situations such as examinations, long-term isolation, and transfers to other hospitals. Furthermore, patients witnessed other patients’ reactions to treatment, some of them extreme. Some patients felt disgusted by the excessive behaviors that they encountered during treatment, either from other COVID-19 patients or from the authorities. Consider the following example:

(6) *Patient 22: ++DISGUST (NN)++ The most unforgettable thing was that after my son and I were discharged from the hospital, another old woman came in. A nurse gave her medicine, and the old woman spat in the nurse’s face. I saw that the nurse was angry, but then she squeezed her hands and was relieved. I really admire the nurse, but I don’t know her name. My most unforgettable memory is about her.*

Excerpt (6) illustrates the lack of cooperation COVID-19 patients sometimes encountered in hospitals. In unfamiliar and stressful environments, some patients might show abnormal behavior patterns. If measures had been taken to address patients’ resistance, medical staff could have worked more efficiently.

COVID-19 patients might encounter unfriendly people, including some medical staff. Inappropriate behavior in patient management may exacerbate COVID-19 patients’ negative emotions, as illustrated in the following example:

(7) *Patient 1: ++DISGUST (NN) ++/++SAD (NB/NJ)++ On the one hand, I sometimes felt a little angry, and then I felt upset and helpless because I didn’t know how to deal with this kind of malice from some officials. I can’t say it in a straightforward manner. In a word, those who exercised their powers might feel uncomfortable with us.*

In excerpt (7), the patient indirectly referred to an undesirable situation in which she was offended. The unclear communication about the nucleic acid test, hospital quarantine, and scientific information of COVID-19 might cause misunderstanding and negative emotions of disgust and sadness among patients. Therefore, new regulations should be established in the healthcare system to treat COVID-19 patients in a more patient-oriented manner.

#### 3.4.3. Misfortune and Suffering

The emotion that preoccupies COVID-19 patients the most is a sense of misfortune and suffering. They constantly question why such a tragedy happened to them. Patients in this study struggled to accept reality. Consider the following example:

(8) *Patient 18: ++FEAR(NE)++/++SAD(NB)++ Um…at that time, my mood was almost collapsed! I didn’t know whether the COVID-19 disease could be cured because my sister demonstrated severe symptoms, and the notice of critical illness was issued. Oh, I was completely collapsed!*

Excerpt (8) vividly describes a patient’s depressive mood and sadness upon learning of their infection. Patient 18 had progressed through the stage of resistance and then experienced extreme negative emotion. Her sister had been suffering from COVID-19 symptoms, too. Having undergone the painful experience of COVID-19 infection, she could hardly accept that her sister was experiencing the same misfortune.

Long hospital quarantine and treatment aggravated patients’ upset emotion and increased their desire to leave the hospital, as described by a man:

(9) *Patient 10: ++SAD(NB)++ Discharged from the quarantine hospital in the Gele Mountains, the ambulance took me home. On the way down the mountain, I felt that this journey was too difficult for me. It was completely different from the usual road I took. I asked the driver, “Why don’t you drive faster?” I really wanted to escape from that position as fast as possible; I really wanted to escape*.

On the day of discharge, Patient 10 was eager to escape from the COVID-19 hospital. In his view, the road leading home was even more distressing than the moment at which he became ill. His isolation in the hospital had caused him great pain and sadness. Once he was discharged, he wished to escape not only the physical location of the hospital but also the psychological trauma that he had suffered from social isolation due to COVID-19.

COVID-19 infection and treatment undoubtfully put enormous stress on patients, as shown in the following extracts:

(10) *a. Patient 29: ++SAD(NB)++ It’s still low, isn’t it? I’ll try to amplify my voice a little. And you turned up the volume, and I’ll try to speak louder. I am not very happy when I talk about my experience of contracting COVID-19. When I feel upset, my voice and tone drop.*


*b. Patient 29: ++SAD(NJ/NB)++/++FEAR(NE)++ Roller coaster, roller coaster…that’s it. My feeling regarding the unstable results of the COVID-19 tests was like falling from the highest position to the lowest position. My mood was just like a roller coaster, to be frank; all of a sudden, I couldn’t stand it in my heart and I could hardly accept the test results.*



*c. Patient 11: ++FEAR(NE)++/++SAD(NB)++ I was abroad, and the landlord would not reduce the rent due to the outbreak of the COVID-19 pandemic because he lost a lot of money for this reason. Then I came back; the COVID-19 treatment cost me 6000–7000 RMB. At that time, I had no money, and I felt like a big mountain was falling on my heart… (very depressed, inaudible).*


Patient 29 discussed the painful emotions that she experienced during her illness for several minutes. At the beginning, the volume of her voice was very low, which she attributed to her upset emotion. Furthermore, she compared her emotional changes with being on a roller coaster as the COVID-19 test results fluctuated. The retest positivity of COVID-19 increased her disappointment, sadness, and depression. For Patient 11, economic hardship was a major risk factor for his depressive and upset emotions. Several other patients also mentioned the economic losses that they had suffered during the pandemic, such as unaffordable rent, business shutdowns, and treatment-related costs. They all exhibited anxiety about whether they could resume their work after recovery.

#### 3.4.4. Discomfort with Physiological Symptoms

Physiological symptoms of COVID-19 infection severely affected patients’ emotional stability. Several examples follow:

(11) *a. Patient 16: ++SAD(NB)++/++DISGUST(ND)++ When I contracted COVID-19, I felt like a big mountain was collapsing on me. The COVID-19 disease came so fiercely like a mountain falling down, but it was a long journey to recover from the virus.*


*b. Patient 18: ++DISGUST(ND)++/++FEAR(NE)++ When I came back from the hospital treatment, I couldn’t sleep well. I experienced insomnia. When I woke up…I turned over and over on the bed; I couldn’t fall sleep anymore.*



*c. Patient 9: ++DISGUST(ND)++ At that time, there were still some physical symptoms. My throat was hoarse and extremely uncomfortable, and I was very likely to feel exhausted when I walked upstairs or climbing. I felt my lungs were obviously hurt by COVID-19. I definitely felt different from normal, healthy people.*


Excerpts (11a–c) show the physical impact of COVID-19 infection, with symptoms including adynamia, insomnia, and anxiety. Such symptoms evoked patients’ sadness, disgust, and fear. This suggests that medical staff should pay close attention to COVID-19 patients’ mental health during physical treatment.

### 3.5. Self-Restriction Due to Probable Contagiousness

Patients were worried about exposing the public to COVID-19. They were likely to be secretive about their own experiences and emotions. They worried that the news about their infection will spread in the community. To forget their past experience of infection and start a new life, COVID-19 patients devoted themselves to work, and they did not play along with the public’s desire for mass entertainment. However, this conservative attitude tended to increase their psychological burden and lead to negative emotions. See the following examples:

(12) *a. Patient 24: ++DISGUST (ND)++ If the news report did not expose my case, no one in my hometown would know about my infection. Since my hometown was in the countryside, people there were talkative about such news regarding my infection. If people in the countryside talk about an event, they might exaggerate the contagiousness of COVID-19 although my symptoms were not severe. Hence, I didn’t want those in the countryside to know about my experience of COVID-19 infection.*


*b. Patient 23: ++FEAR(NE)++/++SAD(NB/NJ)++ We went to the emergency room for a COVID-19 test, and I was unfortunately diagnosed with an infection. People looked at the infected person with strange eyes…I could understand it. When we went to the test window, people intentionally kept a long distance from us across the wall, peering at us. And if we were not following up, they just held their faces, stretched out their heads, and checked whether we could keep up the line. I was afraid of getting close to them, and they were afraid of losing track of us. At that time, I felt…I felt like a…I realized that I was as horrible as the plague. That feeling was…I was a horrible plague…the first feeling was being…um, I was a contagious plague.*


As shown in excerpts (12a–b), people in the community and in the hospital exaggerated the contagiousness of COVID-19. This severely harmed COVID-19 patients’ emotions. Patients felt disgusted, depressed, disappointed, and upset and developed a self-loathing mentality due to their fear of infecting others. Notably, Patient 23 metaphorically regarded himself as a dangerous object, which is strong evidence of self-alienation. In this sense, patients might feel that they should take responsibility for others’ health and that their existence might pose potential health risks for others. These patients’ feelings of personal responsibility made them feel the need to distance themselves from the crowd, leading to negative emotions such as depression, loneliness, and sadness.

### 3.6. Powerlessness about the Disease

The lengthy and frustrating treatment procedures in the hospital destroyed COVID-19 patients’ confidence in their recovery. Consequently, patients might change their attitude toward life, concluding that it was grand but uncontrollable. The preciousness and beauty of life contrasted sharply with the pain of COVID-19. Patients felt fragile and powerless at the hands of the deadly virus, evoking sadness and fear as narrated by Patient 9 and Patient 31:

(13) *a. Patient 9: ++SAD(NB)++/++FEAR(NE/NC)++ Alas…I feel that life is very fragile. Anyway, life is actually very powerful if you regard it as powerful. We survived after struggle in the most difficult pandemic time, which made you feel…ah, life is valuable….But, you fought hard and struggled with COVID-19…after this, there was no way out, so we could only accept the fate, and at this moment, life seemed very fragile.*


*b. Patient 31: ++SAD(NB)++ But, but we often think that this (COVID-19 disease) is eternal, so there will be a lot of pain. We know this truth, but we can’t do it, so there will still be a lot of pain, just like my case now. But, when you are in pain, even if it is short, or when you are sad, even if it is fast and short, you still feel that’s too long.*


Even if patients had recovered from COVID-19′s heavy blow to their health, they still felt upset because they claimed that they could not return to “normal” mentality that they had before being infected. They no longer saw themselves as complete and healthy individuals. COVID-19 infection left a lasting impression, with physiological changes, mental setbacks, and life adjustments, as described in the following extract:

(14) *Patient 3: ++SAD (NJ)++ Because I had the experience of contracting COVID-19…I should be regarded as a disabled person.*

### 3.7. Open-Mindedness about Death

Death was not a sensitive topic among the COVID-19 patients in this study, with almost none directly expressing their fear of and resistance toward death. Some mentioned that the disease had caused their health to deteriorate and brought them to the brink of death, but their fear seemed to concern the disease itself rather than death. When asked about their attitude toward death, patients unanimously voiced that death was not an unacceptable thing, as shown in excerpts (15a–b):

(15) *a. Patient 16: ++HAPPY(PE)++ I think, that is, I haven’t thought so much about death. My mentality about life and death is open, I haven’t thought about it yet. Oh, the view of life and death, namely birth, old age, illness, and death…I haven’t thought about it yet.*


*b. Patient 28: ++HAPPY(PE)++ How should I say this? Life and death are destined. I think it is. If you are destined to die at a certain time, you can’t run away. I view this very naturally, so I don’t consider it right now. If the devil wants to take your life way, you can’t run away, can you? I haven’t thought about these problems yet…I take them easily.*


These two patients’ attitudes toward death reflect the Chinese culture. Traditional Chinese culture emphasizes the impermanence of life, and a valuable death is more meaningful than a dishonorable existence. Hence, these COVID-19 patients did not exhibit fear about death or the end of life. In contrast, they demonstrated positive emotion and felt at ease about future life. They focused on how to live in the present rather than how to prevent themselves from reaching the end. Their calmness about life and death, rooted in the subconscious, was a buffer against the unknown threat of COVID-19.

### 3.8. Faith in the Joint Efforts to Fight COVID-19

Generally, a sense of faith in the joint efforts to fight COVID-19 can positively impact patients’ emotions. Adequate support from fellow patients, friends, colleagues, medical staff, and the government can enhance patients’ positive emotion and reshape their healthy identity [9,10].

#### 3.8.1. Mutual Support between Patients

Social interactions and mutual support between COVID-19 patients play an important role in emotion regulation during treatment [9], as shown in excerpts (16a–c):

(16) *a. Patient 2: ++HAPPY(PB)++ Later, I was transferred to the public health hospital in the Gele Mountains. The woman who lived with me in the same ward is from Xinjiang province. She had no physical discomfort after being confirmed with COVID-19 infection. She danced inside the ward every day, and this relieved our mood.*


*b. Patient 7: ++HAPPY(PE/PB)++ We created a social networking group to communicate during treatment and recovery. We usually sat together and shared the recovery experience face to face. We also shared our life experiences after we were discharged from the hospital. If a peer patient had some symptoms, we could communicate and console each other to relieve the psychological pressure. We were communicating with each other all the time. We shared the COVID-19 test results on a regular basis.*



*c. Patient 18: ++HAPPY(PE)++ My sisters and I chatted via video calls in the social network group every day. Every day! My older sister showed severe symptoms during treatment in the hospital, so I gave her a video call every day, talked to her, and she was very happy. We missed each other if we hadn’t seen each other for a day. Then, at night, my sister didn’t know how to send luck money in the social network group, so I sent luck money and said, “Send luck money with two RMB Yuan, five times. Whoever got the best luck must send luck money.” My older sister got the luck money, and it made her happy. That’s it. We got over the toughest month in the hospital. Haha, every day was the same.*


In these examples, participants communicated their ideas and lived experiences concerning how patients positively influenced each other during treatment in the hospital. They encouraged each other in the journey of recovery and shared knowledge about the virus. Consequently, they could better understand each other and felt safe among fellow patients. The positive interactions between patients inspired hope and happy emotion during their quarantine. Notably, other patients’ recovery positively regulated COVID-19 patients’ emotions. Thus, the beneficial role of recovered fellow COVID-19 patients in psychological interventions should not be underestimated [9].

#### 3.8.2. Inclusive and Considerate Social Environment

Social support from friends and colleagues are both significant in reducing patients’ emotional stress [10], as illustrated in the following excerpts:

(17) *a. Patient 20: ++HAPPY(PB)++ My friends never showed that they were afraid of me or something about COVID-19 infection. Although I didn’t know what they were thinking about, they didn’t talk or behave in other ways to criticize me or isolate me.*

*b. Patient 13: ++HAPPY(PB)++ As for my company, I had been working here since my internship. Later, because I didn’t want to take business trips, I came back. Then, when our company knew about my case of COVID-19 infection, they accepted me. After I went back to the company, I had a high fever, so our company permitted me to work at home. I didn’t go back until around the National Day, when they asked me to return to work as usual*.

Friends and colleagues showed tolerance and kindness toward these two patients. Specifically, friends did not isolate the patients and expose their experience of COVID-19 infection, instead treating them as they would treat ordinary people. In this way, Patient 20′s emotions during recovery became positive. By the same token, understanding and friendliness from colleagues prevented Patient 13 from being stigmatized, allowing for a return to work as a healthy person and fostering positive emotion.

#### 3.8.3. Respect and Tender Love for Medical Staff

COVID-19 patients need support not only from their family members, friends, and colleagues but also from medical staff. Medical staff in the hospital took good care of COVID-19 patients in this study, “warmed their heart,” and reduced their negative emotions [10]. Consider the following examples:

(18) *a. Patient 2: ++HAPPY(PH/PB)++ I think frontline healthcare workers are really like the sun; that is, every time they came into our ward, they were shining with great energy and hope. They brought us hope and encouragement.*


*b. Patient 32: ++HAPPY(PH/PD)++That’s when the COVID-19 pandemic broke out in Wuhan last year. The medical staff were brave fighters carrying arms. They were charging on the frontlines to protect people’s life and health.*



*c Patient 22: ++HAPPY(PH/PB)++ I was in tears when I was discharged from the hospital. If it weren’t for the doctors and nurse, I might not have been able to get cured. They were like a mother; only a mother would do her best to save her children.*



*d. Patient 2: ++HAPPY(PH/PB)++ Yes, I used to think they were very powerful. In fact, after taking off their protective clothing, the nurses were just little girls. They were able to do it seriously, and they were so persistent and so kind to us. Then, we thought they were really great and unselfish.*



*e. Patient 18: ++HAPPY(PH/PG)++ Well done, well done really. The nurses and doctors never discriminated against us, and they worked in the isolation ward with no complaints. I admire them so much! To tell you the truth, they are all little girls and boys. They are only in their teens and twenties, and they are all as young as my children. To be honest, my heart was still…very painful to watch them stay up late and work hard like that. Awesome!*


In excerpts (18a–e), patients compared medical staff to sunshine, brave fighters, family members, and heroes. These narrations revealed patients’ happy emotions in relation to the medical staff’s great dedication, in particular respect, trust, praise, and love. During isolation in the hospital, COVID-19 patients relied on medical staff’s excellent care. They recovered both physically and mentally due to healthcare workers’ selflessness. Patients admired the staff and cooperated with them during treatment. Thus, patients showed positive emotions during the late stages of recovery. This suggests that medical staff should provide scientific knowledge about treatment and counseling for COVID-19 patients, which can majorly improve patients’ emotions [9].

#### 3.8.4. Trust and Love for the Country

The government’s fight against COVID-19 can positively impact patients’ emotions as well. The Chinese government has worked tirelessly to establish policies to combat the COVID-19 pandemic. In this environment, hospitals, communities, schools, and companies have been operating effectively, enhancing COVID-19 patients’ sense of security. Consider the following examples:

(19) *a. Patient 20: ++HAPPY(PG/PH/PB)++ China is great. It is good to have a strong motherland. During the COVID-19 pandemic, China has done the best around the world. When I was infected in Nepal, nobody cared about me. Although we needed to be quarantined in the hospital in Nepal, my friend and I weren’t cared for. However, you can see that pandemic control and patient care has been very good in China. I feel proud to be Chinese.*


*b. Patient 10: +HAPPY(PH/PB)++ Heroic country. China is really a heroic country. So is Wuhan. It is really a heroic city. Wuhan has won the battle against COVID-19. After the city was unlocked, I was really excited that day. By that day, Wuhan had won, and I could go back to Wuhan soon.*



*c. Patient 33: ++HAPPY(PG/PH)++ Our country is like a giant family. We are small families. Our country maintains a great sense of security for us.*


As shown in excerpts (19a–c), COVID-19 patients had positive perceptions of the government’s efforts against COVID-19. In their view, the government acted heroically, showing great power in controlling the pandemic. Patients felt proud to be Chinese and were eager to return to China when they had been infected abroad. The effective measures taken by the Chinese government increased patients’ positive emotions of trust, praise, and love.

#### 3.8.5. Positive Feelings of Giving Back to Society

COVID-19 patients were willing to contribute to fighting COVID-19 after their recovery. They felt that society was a united whole. They were willing to undertake responsibility to help those in need, as shown in the following excerpts:

(20) *a. Patient 7: ++HAPPY(PH/PB)++ Then, I felt that I was very moved at that time. When I was isolated in my home, I got to know that the medical equipment in the public health center was in short supply, such as masks. Ordinary people like us could not help in providing important things, so I bought 1000 RMB Yuan worth of masks to donate.*


*b. Patient 13: ++HAPPY(PE)++ What I thought the most? Er…I wanted to come back and live a good life. I discussed this with other patients in the isolation ward at that time—that is, we must do more to contribute to society when we get out.*



*c. Patient 19: ++HAPPY(PH/PA)++ Last time when I came out, I said I wanted to donate blood. I said, “No problem. If my health allows, I would like to donate blood and help others.” Other people had helped me and saved my life. Why couldn’t I help others? This is an obligatory thing, don’t you think so? Our country and society were so kind to me. Others had helped me, and I am willing to help others.*


These recovered patients exhibited happy emotions considering the adequate support they received during medical treatment. They indicated that they were willing to play an active role in fighting COVID-19, such as donating masks, giving blood, and encouraging hospitalized patients. The country’s collective efforts motivated the recovered patients to engage in COVID-19-related affairs. This sense of moral responsibility increased patients’ positive emotions.

## 4. Discussion

The present study used qualitative methods to explore COVID-19 patients’ emotional discourse, focusing on their lived experiences concerning infection, diagnosis, treatment in isolation, and normal life after discharge. The results showed that emotional frustration poses additional health risks to those infected with COVID-19. In a highly modernized and complex social environment, the psychological problems caused by COVID-19 infection should be treated just as seriously as the disease itself. The findings were interpreted in light of COVID-19 patients’ emotional experiences, mental health problems relating to COVID-19 infection, as well as risk factors and implications for practical solutions.

### 4.1. Patients’ Emotional Experiences

COVID-19 patients’ emotions fluctuated over time. On receiving their diagnosis, they were more likely to feel resistance and suspicion. During the treatment process, depression caused by isolation came to predominate. Their patience for loneliness gradually decreased, and their feelings of attachment to their families increased. After adapting to the isolated environment, patients reflected on the greatness of human life and the temporariness of disease, keeping an open mind about death. During the rehabilitation process, patients were supported by medical staff and other patients, their emotions turned positive, and they began to look forward to a more purposeful new life. After discharge, patients still experienced stigmatization and discrimination. Generally, patients were impressed by the care and support from medical staff, family members, and friends. Their confidence in the joint efforts to fight the COVID-19 pandemic helped them regulate their emotions more effectively [15].

Infection is the first stage of COVID-19 and tends to cause emotional shock. Initially, the bad news of COVID-19 infection was difficult for patients to accept. Patients searched their memories to determine the exact time and place of their infection. Subsequently, patients underwent complex mental changes during treatment. Fear and sadness arose when they felt disturbed by the treatment environment, the uncertainty of the situation caused by the virus, social distance, and discrimination. These negative emotions led to a remarkable increase in loneliness, which worsened in cases of strong attachment to loved ones. Similarly, Jesmi et al. [19] reported that many COVID-19 patients became overly dependent on others. Further, patients quarantined due to possible infection believed that their social relations were stalled by physical restrictions [20].

Pandemic control involves many administrative units and medical institutions. In the hospital, patients might feel that they have been treated unfairly by individual staff members. The patients in the present study complained that the pandemic prevention policy lacked humanistic care. Furthermore, the management regulations of different departments were vague or even inconsistent, causing confusion among COVID-19 patients. This supports previous findings that patients who were isolated at home during the MERS pandemic were irritated by disorganized pandemic prevention measures and that authorities’ credibility was damaged under such circumstances [16,34].

In our study, some patients’ private information was at risk of being leaked, which greatly harmed patients’ emotions. Rates of disclosure of personal information during quarantine have not previously been sufficiently reported in some countries [15,19,21]. However, researchers from regions of East Asia have confirmed similar experiences in COVID-19 patients [16]. This might be related to cross-national differences in cultures or privacy protection regulations. Within high-context societies such as China, information disclosure is often for the purpose of maintaining social well-being.

When patients had adapted to isolation, they were still troubled, but they had begun to look forward to a new life [15]. Patients realized the value of life in their reflections during isolation. They were determined to change their previous attitude about life and to make their future life more meaningful.

Patients’ attitudes toward disease and death reflected traditional Chinese culture. During isolation, patients developed a new understanding of the greatness and fragility of life. Among those who have not experienced hardship, life is a matter of course. However, infectious diseases are often beyond human control. To some extent, patients feel that they have lost part of their lives. Nevertheless, patients in this study generally did not fear death. However, our findings do not support those of Son et al. [16] and Jesmi et al. [19], who reported that a large proportion of patients expressed a fear of death. Fear accounted for only a small proportion of the patients’ emotions toward death in our study due to the influence of traditional Chinese culture. Although patients were worried about the impermanence of life, they did not seem to worry needlessly about death, which positively affected their emotional regulation.

In normal daily interactions, COVID-19 patients receive care and support from individuals around them, such as other patients, friends, and colleagues, aiding their recovery and helping them maintain positive emotions [13]. In their communication with medical staff, patients sensed a professional attitude and care, increasing their confidence in that their health will be restored. Establishing full trust in medical staff can help patients relieve emotional stress [35]. Patients also felt strongly supported by government countermeasures to COVID-19, which enhanced their sense of security. Overall, patients identified themselves as important parts of society, which positively impacted their emotional health.

### 4.2. Mental Health Problems Related to COVID-19 Infection

In unfamiliar, high-pressure environments, COVID-19 infected individuals may exhibit the symptoms of irritation and grief, as well as extreme behaviors when they were treated unfairly. Others’ misbehavior can also make patients feel resentful [22]. In the present study, the strongest negative emotion for COVID-19 patients was suffering over their misfortune, which evoked psychological problems such as anxiety, excessive worries, and stress [9]. In addition, physiological discomfort affects patients’ mental state, as does economic hardship.

Most of the patients’ fear can be attributed to a lack of knowledge about COVID-19. The present study reveals that COVID-19 patients felt great uncertainty because of the limited information they had after diagnosis [16]. This corroborates Sun et al.’s finding that COVID-19 patients felt uneasy and uncertain about the results of medical examinations during their hospitalization [15]. A combination of low mood, uncertainty, and uneasiness may lead to mental health problems such as depression, anxiety, and panic disorder [16,36], consistent with our observations.

Stigma is difficult to eliminate. After leaving the hospital, patients retain the attribute of “disease.” Participants felt rejected by others in society just because they happened to contract COVID-19. To avoid discrimination, patients often refrained from communicating with others. This may result in a higher prevalence of self-perceived post-traumatic stress symptoms among COVID-19 infected individuals [9,24]. Bhattacharya et al. [22] proposed one explanation for this stigma: People are prone to attribute the virus’s spread to others’ negligence due to limited knowledge about the disease and its treatment. Another possible explanation for the social stigma is that isolation, as a widely recognized pandemic prevention measure, inevitably portrays patients as “others” in the hospital, as opposed to “us” outside the hospital [23].

Furthermore, patients in this study were worried that they would pose health risks to others, so they deliberately reduced their social activity. These self-restrictions increased patients’ emotional and mental health burden. These stigmatizing experiences and fears of being stigmatized pose major mental health risks to patients, potentially leading them to make extreme life decisions [22].

### 4.3. Risk Factors and Solutions

In the present study, a lack of scientific information and long-term social isolation constituted the main risk factors leading to COVID-19 patients’ mental health problems. Therefore, one of the priorities of pandemic prevention organizations and healthcare systems should be to provide COVID-19 patients with reliable information and daily emotional support. To alleviate the loneliness and helplessness of patients in isolation, emotional support devices and arrangements, such as daylight windows and activity spaces, should be added in isolation wards [20].

While the Chinese disinfection movement has been rated as the most successful in the world, there are still concerns about the excessive disclosure of personal information due to virus tracking and travel monitoring. Hence, the establishment of a systematic standard to guide the management of private information in epidemiological investigations is urgent [16]. Doing so could help mitigate COVID-19 patients’ anxiety.

Social stigma of certain diseases can cause additional psychological problems for patients. Guidelines from UNICEF, WHO, and IRFC aimed for universal prevention by encouraging trust in the healthcare system and avoidance of language with negative connotations that could lead to stigmatization [37]. Strategies such as educating the public about disease and provision of health information can help reduce stigmatization of patients. The media can also shape public attitudes toward COVID-19 patients [38]. Thus, media messaging campaigns should aim to convey a non-discriminatory tone with respect to patients and the disease. This may prevent the spread of emotional and mental health issues due to stigmatization. Social support from family can also reduce COVID-19 patients’ self-stigma, either from partners or from children, who can act as informal caregivers. “Home” functions as a private and safe place far from public stigma, and it is a spatial symbol of being cured and becoming a “normal” person [18]. The community should increase support for patients’ families and promote mutual care among family members. Moreover, it is essential to create a non-discriminatory atmosphere within the community to reduce the hostility of community members toward patients [26]. The community can guide harmonious exchanges among residents, helping patients eliminate shame and resume a normal social life.

Social confusion in crisis management leads to tension, which may erode trust in the government and hinder the successful implementation of pandemic prevention policies [16]. Therefore, prevention guidelines must be consistent and clear. Authorities should improve their strategies based on the personal experiences of patients, healthcare workers, and other pandemic prevention practitioners. Moreover, more attention should be devoted to educating and training frontline healthcare workers, who play an essential role in helping COVID-19 patients overcome psychological problems [16].

### 4.4. Study Limitations and Future Research

This study had some limitations. First, the themes present are as important as the themes that are absent. We only reported the prominent themes concerning the emotional experiences of COVID-19 patients, while some important themes, such as employment issues due to COVID-19 infection and gender variation in mental health of COVID-19 patients, were underestimated. Second, five basic emotions and the 21 subcategories were used as labels for emotional narrative tagging. However, the 21 subcategories were unequally divided among the five categories, namely seven “HAPPY”, one “ANGRY”, four “SAD”, six “FEAR”, and three “DISGUST”. This might have resulted in potential bias, in particular more themes related to “HAPPY” and “FEAR” being discovered than the themes of “ANGRY” and “DISGUST”.

In the future study, less prominent emotional themes in COVID-19 patient narratives should be reported to reveal the unknown side of patients’ lived experience. Furthermore, the ID Tags of five basic emotions and 21 subcategories need to be balanced and the potential bias should be minimalized in data analysis. In addition, an emotional analysis of patient experience may involve the quantitative analysis of the emotional words. Essentially, quantitative and qualitative measurements could be combined to uncover the emotional and mental health status of COVID-19 patients in a comprehensive way.

## 5. Conclusions

COVID-19 patients’ emotions are dynamic. The eight themes of patients’ emotional narratives identified in the present study mainly concerned the experience of infection, isolation, attitudes toward life and death, stigma, and macro-identity. The results confirmed that during the pandemic, COVID-19 patients were plagued by anxiety, depression, stress, and insomnia because of their isolation. In contrast, adequate social support can enhance patients’ positive emotions. Thus, fully understanding the psychological setbacks experienced by patients during treatment will be helpful for the authorities in improving interventions.

In practice, the government should establish policies to mitigate discrimination and protect COVID-19 patients’ privacy [9,10]. Humane care should be available to all COVID-19-infected persons. Medical staff should not only support patients’ physical recovery but also pay attention to patients’ psychological status [9]. They should also provide scientific knowledge about COVID-19 and treatments to patients. Further, the media should respect COVID-19 patients and should not present a distorted image of them to the public. Taken together, patients’ emotional status can be protected and regulated through joint efforts of governments, medical staff, the media, and all citizens.

## Figures and Tables

**Figure 1 ijerph-19-09491-f001:**
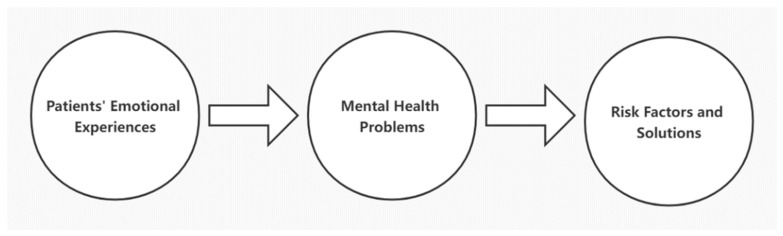
Schematic of the conceptual framework.

**Figure 2 ijerph-19-09491-f002:**
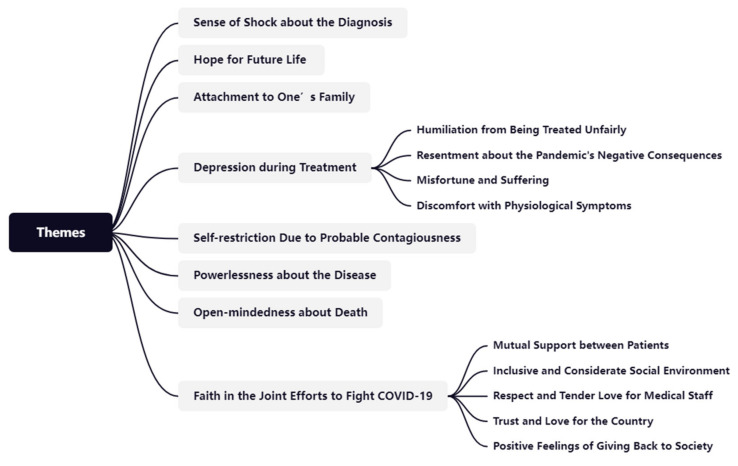
Themes and subthemes of COVID-19 patients’ emotional discourse.

**Table 1 ijerph-19-09491-t001:** Demographic information of the 34 COVID-19 patients.

Patient ID	Sex	Age	Marital Status	Occupation	Quarantine Time (Days)
1	F	21	Single	Student	11
2	F	24	Married	Communication designer	12
3	F	25	Single	Guide doctor	25
4	M	46	Divorced	Carpenter	7
5	F	46	Married	Businessman	30
6	M	30	Married	Interior designer	18
7	F	35	Married	Self-employed	47
8	F	40	Married	Waiter	10
9	M	31	Married	Computer engineer	16
10	M	35	Married	Self-employed	20
11	M	39	Married	Self-employed	17
12	F	48	Married	—	11
13	F	42	Married	Project budget manager	5
14	M	40	Married	Enterprise manager	69
15	M	40	Married	Factory worker	10
16	M	45	Married	Farmer	12
17	M	33	Married	Entertainer	30
18	F	47	Married	Housewife	10
19	F	57	Married	—	34
20	F	51	Married	Self-employed	7
21	M	31	Single	Freelancer	>10
22	F	51	Married	Farmer	35
23	M	34	Married	—	18
24	M	55	Married	Teacher	14
25	M	41	Married	Engineer	7
26	F	51	Married	Farmer	>10
27	M	53	Married	Farmer	18
28	M	38	Married	Surveying engineer	5
29	F	50	Divorced	Retired	16
30	F	29	Single	Student	2
31	F	29	Single	Self-employed	6
32	M	30	Single	Electrical engineer	5
33	M	26	Single	Worker	5
34	M	27	Single	E-commerce	4

**Table 2 ijerph-19-09491-t002:** Emotional framework and tags.

No	Xu et al.’s [28] Basic Emotions	Xu et al.’s [28] Emotional Subcategories	Hong et al.’s [29] Basic Emotions	Polarity
1	Happy	Joy (PA)	HAPPY	Positive
2		Comfort (PE)	HAPPY	Positive
3	Good	Respect (PD)	HAPPY	Positive
4		Praise (PH)	HAPPY	Positive
5		Trust (PG)	HAPPY	Positive
6		Like (PB)	HAPPY	Positive
7		Wish (PK)	HAPPY	Positive
8	Angry	Angry (NA)	ANGRY	Negative
9	Sad	Upset (NB)	SAD	Negative
10		Disappointed (NJ)	SAD	Negative
11		Guilty (NH)	SAD	Negative
12		Grief (PF)	SAD	Negative
13	Fear	Panic (NI)	FEAR	Negative
14		Dread (NC)	FEAR	Negative
15		Shame (NG)	FEAR	Negative
16	Disgust	Depressed (NE)	FEAR	Negative
17		Hate (ND)	DISGUST	Negative
18		Criticize (NN)	DISGUST	Negative
19		Envious (NK)	DISGUST	Negative
20		Suspect (NL)	FEAR	Negative
21	Surprise	Surprise (PC)	FEAR/HAPPY	Negative/Positive

**Table 3 ijerph-19-09491-t003:** Examples of emotional narrative tagging.

Primary Class [29]	Secondary Class [28]	Examples of Emotional Narratives
HAPPY	Joy (PA)	++HAPPY (PA)++ When I was informed that I met the criteria of discharge from the hospital, I felt very excited.
HAPPY	Comfort (PE)	++HAPPY (PE)++ For living, we should keep an open and optimistic view about life and disease.
HAPPY	Respect (PD)	++HAPPY (PH/PD)++ As for the doctors, some of them are very polite and warmhearted to COVID-19 patients.
HAPPY	Praise (PH)	++HAPPY(PH)++ Ah, the nation is like a great mountain, we feel safe with it.
HAPPY	Trust (PG)	++HAPPY(PG)++ We encouraged ourselves. We believed that we can defeat COVID-19. That’s what I was thinking of.
HAPPY	Like (PB)	++HAPPY(PH/PB)++ During the COVID-19 pandemic, many people are showing love to each other and taking care of each other.
HAPPY	Wish (PK)	++HAPPY(PK)++ The first thought regarding recovery from COVID-19 was that I wish that I could leave the hospital as soon as possible and return home.
ANGRY	Angry (NA)	++ANGRY(NA)++ I even quarreled furiously with the medical staff when they informed me of collecting the blood sample for anti-virus research
SAD	Upset (NB)	++SAD(NB)++ During my reinfection, I felt reluctant to talk; I did not want to talk with others. I felt upset and wanted to be by myself.
SAD	Disappointed (NJ)	++SAD(NJ)++ When I found out that I tested positive for COVID-19, I felt disappointed.
SAD	Guilty (NH)	++SAD(NH)++ I had to stay in the hospital; during the hospital quarantine, I felt reluctant to bother others, and I did not want to make troubles (sighed)…
SAD	Grief (PF)	++SAD (NB/PF)++ When I was quarantined in the hospital after confirmed infection, I missed my family very deeply.
FEAR	Panic (NI)	++FEAR(NI)++ When my COVID-19 infection was confirmed, I felt worried, dismayed, and anxious.
FEAR	Dread (NC)	++FEAR(NC)++ During the outbreak of the COVID-19 pandemic in Wuhan, it was unforgettable every day because it was so horrible.
FEAR	Shame (NG)	++FEAR (NG)++ I dare not mention my experience of COVID-19 infection. I have nothing to say about it. It’s a shame to mention it.
FEAR	Depressed (NE)	++ FEAR (NE)++ When I was hospitalized after confirmed infection, I felt gloomy and depressed during the hospital quarantine. It’s a normal response.
DISGUST	Hate (ND)	++DISGUST(ND/NN)++ The most irritating thing is that when I was infected with COVID-19, people in my community regarded me as an infectious monster. They are coldhearted people.
DISGUST	Criticize (NN)	++DISGUST (NN)++ When I was infected with COVID-19, some people even showed discrimination toward patients like us.
DISGUST	Envious (NK)	N/A
FEAR	Suspect (NL)	++FEAR(NL)++ I could not believe that I contracted COVID-19. Why me? I could not believe it. That’s the feeling I had at that time.
FEAR	Surprise (PC)	++FEAR (PC)++ I had no knowledge about COVID-19. I have heard that COVID-19 is dangerous, but I have never thought it was so deadly and infectious until I was infected.

## Data Availability

The data presented in this study are available on request from the corresponding author. The data are not publicly available due to privacy reasons.

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
