# Peer review of "Emotional Experiences of COVID-19 Patients in China: A Qualitative Study"

_ijerph, 2022, doi:10.3390/ijerph19159491_

Round 1

Reviewer 1 Report

Thank you for sending a very interesting manuscript. The topic taken up by the authors is very important. Congratulations on conducting the research.

The article is very well described. The methodological part of the paper is described in a satisfactory way. The whole inference is very reliably presented and does not raise any objections. The fact of quoting the respondents' statements and creating their own model of feeling emotions (fig.1- line 139) is noteworthy.

In my opinion the discussion is very good and comprehensive, and fig.2 which presents the research procedure can be used by other people who would like to conduct similar research.

In my opinion the only objection is too little literature referred to by the authors in the text, and also the introduction should be more elaborated. It would be useful to refer to other similar studies, not only in the context of quality of life and covid- 19- ual disease, but also to studies talking about the assessment of quality of life in relation to health- e.g.

·        Ziółkowska-Weiss, K. Assessment of the Selected Health Factors by Polonia in the Greater Toronto Area in the Context of Quality and Standard of Living, International Journal of Environmental Research and Public Health. 2021; 18(3):1296. 1-20, https://doi.org/10.3390/ijerph18031296

·        Wei-Wei, D.; Garcia, A. Later Life Migration: Sociocultural Adaptation and Changes in Quality of Life at Settlement Among Recent Older Chinese Immigrants in Canada. Act. Adapt. Aging 2015, 39, 214-242.

It may be worthwhile to add a paragraph in the introduction stating that many authors have studied the health-related aspect of quality of life-and here cite several authors who have conducted such studies, and then go on to the section stating that the topic of quality of life and past Covid-19 disease is still unexplored, so the authors fill in the research gap .

Author Response

Dear Reviewer,

We appreciate the interest that you have taken in our manuscript entitled “Emotional Experiences of COVID-19 Patients in China: A Qualitative Study” (Manuscript ID: ijerph-1794421). The comments are very insightful for improving our manuscript. We have made revisions in accordance with your enlightening suggestions. Revised and rewritten portions were marked up using the “Track Changes” function in the revised manuscript. The point-by-point response to your comments is shown below.

Reviewer #1

Thank you for sending a very interesting manuscript. The topic taken up by the authors is very important. Congratulations on conducting the research.

The article is very well described. The methodological part of the paper is described in a satisfactory way. The whole inference is very reliably presented and does not raise any objections. The fact of quoting the respondents' statements and creating their own model of feeling emotions (fig.1- line 139) is noteworthy.

In my opinion the discussion is very good and comprehensive, and fig.2 which presents the research procedure can be used by other people who would like to conduct similar research.

Response: We are grateful to the reviewer for the positive comments.

In my opinion the only objection is too little literature referred to by the authors in the text, and also the introduction should be more elaborated. It would be useful to refer to other similar studies, not only in the context of quality of life and covid- 19- ual disease, but also to studies talking about the assessment of quality of life in relation to health- e.g.

  • Ziółkowska-Weiss, K. Assessment of the Selected Health Factors by Polonia in the Greater Toronto Area in the Context of Quality and Standard of Living, International Journal of Environmental Research and Public Health. 2021; 18(3):1296. 1-20, https://doi.org/10.3390/ijerph18031296
  • Wei-Wei, D.; Garcia, A. Later Life Migration: Sociocultural Adaptation and Changes in Quality of Life at Settlement Among Recent Older Chinese Immigrants in Canada. Act. Adapt. Aging 2015, 39, 214-242.

Response: The reviewer is right that more literature on relevant studies should be provided. In the revised version, we firstly added a paragraph regarding health-related quality of life and the aim of the present study at the beginning of Introduction on page 1, which reads as:

“Quality of life is strongly associated with our health status, which means not only the absence of illness, but also a state of physical, mental and social well-being [1-3]. Thus, the assessment of quality of life is linked with physical health, psychological state, social relationships, personal belief, and our relationship to the living environment. Health-related quality of life has been extensively explored in different medical disorders such as heart disease, HIV, lung cancer, and SARS [4-7]. The present study aims to explore health-related quality of life among COVID-19 patients in China, and its domain specific to emotional experiences regarding COVID-19 infection.”【see page 1】

The newly cited references:

  1. Ziółkowska-Weiss, K. Assessment of the selected health factors by Polonia in the greater Toronto area in the context of quality and standard of living. International Journal of Environmental Research and Public Health, 2021; 18(3):1296. 1-20, https://doi.org/10.3390/ijerph18031296
  2. Research on the Menopause: WHO Technical Report. Series 670. WHO Scientific Group: Geneva, Switzerland, 1981.
  3. Wei-Wei, D.; Garcia, A. Later life migration: Sociocultural adaptation and changes in quality of life at settlement among recent older Chinese immigrants in Canada. Adapt. Aging, 2015, 39, 214-242.
  4. Rosenberg K. Health-related quality of life declines for survivors of congenital heart disease. Am J Nurs. 2021, 121(6): 71. doi: 10.1097/01.NAJ.0000753688.38302.b1.
  5. Fleming, C.A., Christiansen, D., Nunes, D., et al., 2004. Health-related quality of life of patients with HIV disease: impact of hepatitis C coinfection. Infect. Dis. 38 (4), 572–578.
  6. Zikos E, Ghislain I, Coens C, Ediebah DE, Sloan E, Quinten C, Koller M, van Meerbeeck JP, Flechtner HH, Stupp R, Pallis A, Czimbalmos A, Sprangers MA, Bottomley A. Health-related quality of life in small-cell lung cancer: a systematic review on reporting of methods and clinical issues in randomised controlled trials. Lancet Oncol. 2014, 15(2):e78-89. doi: 10.1016/S1470-2045(13)70493-5.
  7. Bonanno, G.A., Ho, S.M.Y., Chan, J.C.K., et al., 2008. Psychological resilience and dysfunction among hospitalized survivors of the SARS epidemic in Hong Kong: a latent class approach. Health Psychol. 27 (5), 659.

Furthermore, we have elaborated more literature about the mental health status of COVID-19 patients in China, as shown on page 2:

“…Another survey questionnaire of PTSD, depression, and anxiety scales with 126 quarantined COVID-19 patients in Shenzhen also demonstrated a high rate of psychological distress among the COVID-19 survivors in the early recovery stage [25]. Using a qualitative narrative analysis method, Wu et al. [18] investigated health-related quality of life of sixteen hospitalized COVID-19 survivors in Nanning city. The narrative analysis showed that COVID-19 patients suffered from anxiety, trauma, and self-stigma. Patients’ negative emotions were linked with abnormal social interactions. Drawing on a thematic analysis of narratives of thirteen COVID-19 patients in Wuhan, Li et al. [26] found that patients experienced negative emotions such as confusion, uncertainty, worry, and guilt due to social discrimination and poor financial security. Patients’ positive emotion was concerned with expectations about making up for lost time with family after discharge….”【see page 2】

The newly elaborated references:

  1. Wu, C.; Cheng, J.; Zou, J.; Duan, L.; Campbell, J. E. Health-related quality of life of hospitalized COVID-19 survivors: an initial exploration in Nanning city, China. Social Science & Medicine, 2021, 274, 113748.
  2. Cai, X., Hu, X., Ekumi, I.O., et al. Psychological distress and its correlates among COVID-19 survivors during early convalescence across age groups. Am. J. Geriatr. Psychiatr. 2020, 28 (10), 1030–1039.
  3. Li, T.; Hu, Y.; Xia, L.; Wen, L.; Ren, W.; Xia, W.; Wang, J.; Cai, W.; Chen, L. Psychological experience of patients with confirmed COVID-19 at the initial stage of pandemic in Wuhan, China: A qualitative study. BMC Public Health 2021, 21, 2257. https://doi.org/10.1186/s12889-021-12277-4

It may be worthwhile to add a paragraph in the introduction stating that many authors have studied the health-related aspect of quality of life-and here cite several authors who have conducted such studies, and then go on to the section stating that the topic of quality of life and past Covid-19 disease is still unexplored, so the authors fill in the research gap.

Response: Thanks for this constructive comment. As illustrated in the above points, we have added a whole paragraph on the health-related aspect of quality of life and its assessment at the beginning of the introduction on page 1. Then, we have elaborated literature on the quality of life among Covid-19 patients, in particular psychological well-being of COVID-19 patients in the context of China. Finally, we identified the research gap and significance on pages 2-3: “although there have been several broad qualitative studies concerning COVID-19 patients’ mental health in the context of China, there is a lack of in-depth studies examining the emotional status of patients with confirmed cases of COVID-19……This study can serve to determine the prevalence of certain emotional tendencies in COVID-19 patients and to identify relevant interventions to improve their well-being”.

Thank you for again for your interest and efforts and we are ready to respond to any further questions and comments you may have.

Best wishes,

The authors

Reviewer 2 Report

Dear Authors!

I read your article with great interest and I find the whole work and the qualitative design very suitable and important for the modern medicine and clinic psychology.

I may only advise you the following:

1) add some reasons for choosing qualitative research;

2) explain the table 2 with more details, it is difficult to understand the crosses before and after the words and the capital letters as PH, PD, N and so on.

Please, describe all the signs used in the table. Also, it would be interesting if you stress some predominant type of answers and describe better the whole procedure (all the questions were structured? was it free answers or the choices? how man questions did you use for each subject?)

So, please, make the whole procedure more complete and more clear for the readers. Please just show some dominant or non predominant tendency of the answers. Please, show the strengths of qualitative research  in general and in yours particular case.

Thank you and good fortune!

Reviewer 3 Report

The paper addresses an important subject but not scientifically. Its shortcomings could be corrected but the paper would need a major revision. The weaknesses are as follows:
• The study is presented as more an ad hoc one than an exploratory one. It is framed using the five basic emotions and the 21 subcategories of emotions. But the framework is mentioned in the passing in the methods and materials and neither explained nor supported.
• The 21 subcategories are very unequally divided among the 5 categories – 7 Happy, 1 Angry, 4 Sad, 6 Fear, and 3 Disgust. Would that result in more happy and fear themes being discovered and not angry and disgust ones?
• The instantiations of category-subcategory combinations are not convincing.
• There is very little explanation of the method of extraction of the themes and subthemes. It is left to NVIVO. Please note that among other issues the themes/subthemes could be severely biased by the distance measure and method used in clustering.
• Scientifically, the themes present are as important as the themes that are absent (and could/should have been present theoretically). The latter are not analyzed.
• The presentation of the results is very difficult to decipher. The link(s) between the themes/subthemes and the emotion category/subcategory combinations is not clear.
• The presentation of the results is incoherent.
• The discussion does not follow the results. The emergence of the schematic in Figure 2 is unclear. For example, the concepts of chronology and risk factors are introduced anew.
• Is Figure 2 the theoretical/conceptual framework?
The authors have some rich interview data. They need to be studied more systematically with a systemic theoretical/conceptual framework. The method must be explained in greater detail and its potential biases need to be explored in interpreting the results.

Round 2

Reviewer 3 Report

The authors have responded to my comments adequately.